# Learning Interpretable BEV Based VIO without Deep Neural Networks

**Zexi Chen   Haozhe Du   Xuecheng Xu   Rong Xiong   Yiyi Liao*   Yue Wang**

Zhejiang University

{chenzexi,hzdu,xuechengxu,rxiong,yiyi.liao,ywang24}@zju.edu.cn

**Abstract:** Monocular visual-inertial odometry (VIO) is a critical problem in robotics and autonomous driving. Traditional methods solve this problem based on filtering or optimization. While being fully interpretable, they rely on manual interference and empirical parameter tuning. On the other hand, learning-based approaches allow for end-to-end training but require a large number of training data to learn millions of parameters. However, the non-interpretable and heavy models hinder the generalization ability. In this paper, we propose a fully differentiable, and interpretable, bird-eye-view (BEV) based VIO model for robots with local planar motion that can be trained without deep neural networks. Specifically, we first adopt Unscented Kalman Filter as a differentiable layer to predict the pitch and roll, where the covariance matrices of noise are learned to filter out the noise of the IMU raw data. Second, the refined pitch and roll are adopted to retrieve a gravity-aligned BEV image of each frame using differentiable camera projection. Finally, a differentiable pose estimator is utilized to estimate the remaining 3 DoF poses between the BEV frames: leading to a 5 DoF pose estimation. Our method allows for learning the covariance matrices end-to-end supervised by the pose estimation loss, demonstrating superior performance to empirical baselines. Experimental results on synthetic and real-world datasets demonstrate that our simple approach is competitive with state-of-the-art methods and generalizes well on unseen scenes.

**Keywords:** VIO, Interpretable Learning

## 1 Introduction

Visual-inertial odometry (VIO) is a fundamental component of the visual-inertial simultaneous localization and mapping system with many applications in robotics and autonomous driving. As a cheap and efficient solution, monocular VIO has attracted growing interest recently. However, monocular VIO is a challenging task considering the limited sensory information.

Traditional monocular VIO approaches have demonstrated promising results based on classical feature detection and feature matching [1][2][3][4], see Fig. 1 (top). These methods are interpretable and generalize well, but require manual interference and empirical parameter tuning. Moreover, these methods only leverage sparse features, resulting in the degraded performance in textureless regions [5].

More recently, learning-based approaches address these limitations by learning the ego-motion in an end-to-end fashion from the raw, dense monocular images, as illustrated in Fig. 1 (middle). However, they lack interpretability and thus pose a new challenge to generalization. While many works improve the interpretability by predicting an intermediate optical flow [6][7][8][9] or depth map [10][11][12][13], the following pose regression network remains uninterpretable. Furthermore, existing learning-based approaches rely on a large number of training images to learn millions of parameters while the ego-motion has essentially 6 DoF only. We thus ask the following question: *Is*

---

*co-corresponding author, ** corresponding author.

6th Conference on Robot Learning (CoRL 2022), Auckland, New Zealand.

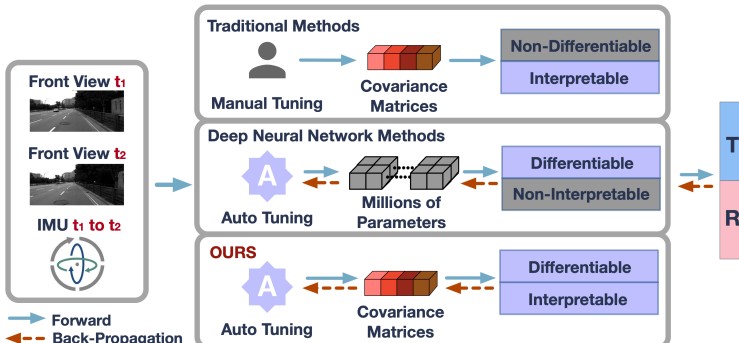

Figure 1: **Comparison of VIO methods**. **Top:** the standard visual-inertial odometry that requires manual tuning parameters, **middle:** Deep learning based VIO approaches learn depth or optical flow prediction before a pose regression, and are loaded with tons of heavy parameters for training, and **bottom:** our differential interpretable model based learning approach that solves the problem in an end to end manner, training covariance matrices without deep neural networks.

*it possible to obtain a fully interpretable model with a minimal set of parameters being learned end-to-end?* In this paper, we focus on the BEV based VIO for 5 DoF local planar motion with elevation fixed. This is not trivial as such motion pattern is applicable for unmanned ground vehicles and autonomous driving vehicles in urban, underground parking lots, tunnels and highways [14][15]. We combine the advantages of the traditional methods with learning-based approaches. Several former works [16][17] explore this possibility using traditional filtering for interpretable fusion, but leaving the sensor model a plain regression network. Our key idea is to design a fully differentiable, interpretable model that eliminates all black-box neural networks, yet allowing for replacing empirical parameter tuning with learning from the data, see Fig. 1 (bottom). Additionally, our method exploits dense appearance information instead of sparse features. As it is difficult to differentiate the 6 DoF pose estimation that involves robust and non-linear optimization, we focus on the 3 DoF image based pose estimation conditioned on the estimated pitch and roll angles.

More formally, we consider the BEV based monocular VIO as a function that outputs a relative pose taking as input i) two consecutive frames, and ii) IMU data. First, we utilize a Differentiable Unscented Kalman Filter (DUKF) to filter out noise from the IMU raw measurements and retrieve pitch and roll. As the covariance matrices of noise in both the motion model and the measurement model are not known precisely in practice, classical UKF relies on empirical parameter tuning given a specific scene. In contrast, we exploit the fact that UKF is differentiable and optimize the covariance matrices in an end-to-end manner leveraging the training data. This requires the estimator of the remaining 3 DoF pose to be differentiable as well. Therefore, in the second step, we estimate the remaining 3 DoF poses by converting monocular images to bird-eye view conditioned on the denoised pitch and roll [14]. The bird-eye view representation allows us to obtain a global optimal solution using Differentiable Phase Correlation (DPC) [18] in a single forward pass. Note that DPC leverages dense appearance information and does not introduce any trainable parameters. More importantly, it allows for back-propagating the gradient to the DUKF module to update the covariance matrices. The combination of DUKF and DPC provides us a fully differentiable, and interpretable BEV based model for VIO, named BEVO, with the parameters of the covariance matrices as the *only* trainable parameters and no deep neural networks involved. We emphasize that by carefully combining the designed mechanistic model with learning approaches, the model is inherently interpretable. [19]. We summarize our contributions as follows:

- We present a fully differentiable and interpretable monocular VIO framework for planar moving robots, e.g., autonomous driving or unmanned ground vehicles, that combines the advantages of both traditional and learning-based approaches.

- We propose to use the Unscented Kalman Filter as a differentiable layer and leverage the Differentiable Phase Correlation on BEV images for pose estimation. This novel combination allows for learning the covariance matrices from data in an end-to-end manner.

- Extensive experiments on KITTI, CARLA, and AeroGround demonstrate that, surprisingly, our simple approach shows competitive results compared to the state-of-the-art.

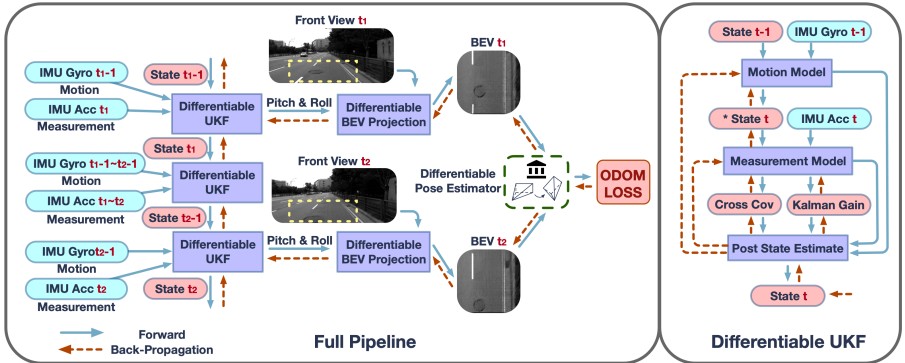

Figure 2: The **overall pipeline** of the proposed BEVO. As a demonstration, the frequency of the front image is 10Hz while the frequency of IMU is 100Hz. For each frame of IMU, the pitch rate, roll rate, and 3D acceleration are fed into the differentiable UKF to learn noise filtering (elaborated in the right part of the figure). Afterward, the filtered pitch and roll aid the differentiable BEV projections which are further estimated for a pose to be supervised. However, due to the frequency difference, only frames with front images have back-propagated gradients (red dashed lines) for the Differentiable UKF (DUKF) training.

## 2 Fully Differentiable and Interpretable Model for VIO

We aim to estimate the odometry with the input of one monocular camera and IMU data by a differentiable and interpretable model. Following the pipeline in Fig. 2, we employ a trainable UKF to integrate IMU data for pitch and roll estimation. Then the denoised pitch and roll, together with the corresponding front image are fed into the second differentiable module, for differentiable pose estimation of the remaining 3 DoF pose. To this goal, we transform the image to the BEV images conditioned on the denoised pitch and roll. Following the BEV projection, the $\mathbb{SIM}(2)$ pose of two consecutive BEV images is estimated by the DPC. Note that the UKF is a continuous forward function and is differentiable in essence. In this work, we emphasize its backward path by referring to it as DUKF, to distinguish it from the traditional UKF utilization which focuses only on the forward path. As a result, the only trainable parameters are the 4 numbers that make up the covariance matrices of motion and measurement in the DUKF and are trained end-to-end with the pose loss altogether.

### 2.1 Differentiable Unscented Kalman Filter

For the DUKF, there are two basic models in the pipeline, namely, the motion model and the measurement model. We follow the right-hand system through the paper.

**Motion Model:** Given the filtered pitch and roll in time $t - 1$, we have the state in time $t - 1$ as:

$$x_{t-1} = [\alpha_{t-1}, \quad \beta_{t-1}]^T, \tag{1}$$

where $\alpha$, and $\beta$ are the pitch and roll angle respectively. Following the sampling theory of UKF [20], the sampled state in $t - 1$ becomes $X_{t-1} = \Gamma(x_{t-1})$, where $\Gamma$ is the sampling rule. Given the motion model $G$, we have the initial guess of pitch and roll in time $t$ as:

$$\bar{X}_t^* = G(X_{t-1}) = [\alpha_{t-1} + \omega_{\alpha_{t-1}} \Delta t, \quad \beta_{t-1} + \omega_{\beta_{t-1}} \Delta t]^T \tag{2}$$

$$\bar{\mu}_t = \sum w_m \bar{X}_t^*, \quad \bar{\sigma}_t = \sum w_c (\bar{X}_t^* - \bar{\mu}_t)(\bar{X}_t^* - \bar{\mu}_t)^T + \boldsymbol{O_t}, \tag{3}$$

where $\omega_\alpha$, and $\omega_\beta$ are the pitch and roll rate from the gyroscope, and $\bar{\mu}_t$ and $\bar{\sigma}_t$ are the mean and variance of the initial guess of $\bar{X}_t^*$, respectively. $\Delta t$ is the interval of the IMU data. Note that $w_m$ and $w_c$ are the weights for each sampled state in $\bar{X}_t^*$ based on the theory in [20], and $\boldsymbol{O_t}$ is the covariance matrix for the noise of the motion model to be trained.

**Measurement Model:** After updating the covariance matrix $\boldsymbol{O_t}$ which is the motion prediction, we retrieve a new sampled state $\bar{X}_t$ in time $t$:

$$\bar{x}_t \sim N(\bar{\mu}_t, \bar{\sigma}_t) \triangleq [\bar{\alpha}_t, \quad \bar{\beta}_t]^T \tag{4}$$

$$\bar{X}_t = \Gamma(\bar{x}_t), \tag{5}$$

where $\bar{x}_t$ satisfies the normal distribution. Given the measurement model $H$, the initial guess of measurement is

$$\bar{Z}_t = H(\bar{X}_t) = \begin{bmatrix} \bar{\alpha}_t, & \bar{\beta}_t \end{bmatrix}^T \tag{6}$$

$$\bar{M}_t = \sum w_m \bar{Z}_t \tag{7}$$

$$\bar{\Sigma}_t = \sum w_c (\bar{Z}_t - \bar{M}_t)(\bar{Z}_t - \bar{M}_t)^T + \boldsymbol{Q_t}, \tag{8}$$

where $\bar{M}_t$ and $\bar{\Sigma}_t$ are the mean and variance of $\bar{X}_t$ respectively, and $\boldsymbol{Q_t}$ is the covariance matrix for the noise of the measurement model to be trained.

Now, considering a real-world measurement is obtained:

$$\hat{\alpha}_t = -arctan(acc_t^x/\sqrt{acc_t^{y2} + acc_t^{z2}}), \quad \hat{\beta}_t = arctan(acc_t^y/acc_t^z), \quad Z_t = \begin{bmatrix} \hat{\alpha}_t, \hat{\beta}_t \end{bmatrix}^T, \quad (9)$$

where $\hat{\alpha}_t$ and $\hat{\beta}_t$ are the measurement of pitch and roll in time $t$ with respect to the raw acceleration data in each axes $acc_t^x$, $acc_t^y$, and $acc_t^z$. Altogether, the real-world measurement state is formed as $Z_t$.

By now, we come to the last stage of the DUKF, update the final state $x_t$ as follows:

$$\bar{\Sigma}_t^{X,Z} = \sum_{i=0}^{2n} w_c{}^i (\bar{X}_t^i - \bar{\mu}_t)(\bar{Z}_t^i - \bar{M}_t)^T \tag{10}$$

$$K_t = \bar{\Sigma}_t^{X,Z} \bar{\Sigma}_t^{-1} \tag{11}$$

$$\mu_t = \bar{\mu}_t + K_t(Z_t - \bar{M}_t), \qquad \sigma_t = \bar{\sigma}_t + K_t \bar{\Sigma}_t K_t{}^T \tag{12}$$

$$x_t \sim N(\mu_t, \sigma_t) \triangleq \begin{bmatrix} \alpha_t, & \beta_t \end{bmatrix}^T, \tag{13}$$

where $\bar{\Sigma}_t^{X,Z}$, $K_t$ are the cross-covariance and Kalman Gain respectively, $\mu_t$ and $\sigma_t$ are the mean and variance of $x_t$ which will also guide the sampling in the following time $t+1$.

## 2.2 Differentiable Bird-eye View Projection

To achieve the projection from the front view to the birds-eye view with DBEV, we first define the transformation matrix from the IMU to the ground beneath the vehicle considering its pitch, roll, and elevation:

$$R_t^{IMU} = \begin{bmatrix} cos\alpha_t & 0 & -sin\alpha_t \\ sin\alpha_t sin\beta_t & cos\beta_t & cos\alpha_t sin\beta_t \\ sin\alpha_t cos\beta_t & -sin\beta_t & cos\alpha_t cos\beta_t \end{bmatrix}, \quad T_t^{IMU} = \begin{bmatrix} 0 & 0 & const_z \end{bmatrix}, \tag{14}$$

the rotation matrix $R_t^{IMU}$ of the IMU at time $t$ changes constantly with $\alpha_t$ and $\beta_t$ which are estimated by the DUKF. The translation vector $T_t^{IMU}$ of the IMU is known and fixed since it is the urban road and highway that we are considering in which the height of the IMU to the ground $const_z$ stays unchanged through the journey [14]. Transformation matrices from camera to the world $[R_t^{CAM}, T_t^{CAM}]$ are obtained with the calibration from IMU to camera and $[R_t^{IMU}, T_t^{IMU}]$.

Besides the unchanged height, one more assumption can be made for a safe trip: there should be sufficient distance ahead which should be about $10m$ at $34km/h$ according to the mounting height and angle (similar to [14] [15]), forming that the center bottom part of the front image should represent a plane ground, shown as the yellow dashed box in Fig. 2.

With the assumption above, we consider pixels within the yellow dashed box as a set of points in the 3D space, and map them from the camera plane to the ground plane.

To begin with, we generate a set of even points $P_g$ on the ground plane that stands for the location that the points $\{u, v\}$ in the BEV should be projected to:

$$P_g \triangleq \{X^{'}, Y^{'}, Z^{'}\}^T, \tag{15}$$

where $X^{'}$, $Y^{'}$, and $Z^{'}$ are the coordinates of points. $P_g$ is then transformed to the camera coordinate with:

$$P_c^t = R_t^{CAM^T}(P_g - T_t^{CAM}) \triangleq \{X_t^{''}, Y_t^{''}, Z_t^{''}\}^T, \tag{16}$$

and $P_c^t$ is projected to the camera plane with the focal lengths $f_x$, $f_y$ and principal point $(c_x, c_y)$:

$$Z_t'' \begin{bmatrix} u_t' \\ v_t' \\ 1 \end{bmatrix} = \begin{bmatrix} f_x & 0 & c_x \\ 0 & f_y & c_y \\ 0 & 0 & 1 \end{bmatrix} \begin{bmatrix} X_t'' \\ Y_t'' \\ Z_t'' \end{bmatrix}, \qquad (17)$$

where $(u_t', v_t')$ are the indexes on the front camera image of each corresponding point in $P_g$ in time $t$. Therefore, by remapping the original pixels $(u, v)$ to $(u_t', v_t')$, we retrieve the BEV projection $I_t^{bev}$ out of the yellow dashed box $I_t^f$ in Fig. 2:

$$I_t^{bev}(\{u_t, v_t\}) = I_t^f(F(\{u_t, v_t\})), \qquad (18)$$

where $\{u_t', v_t'\} = F(\{u_t, v_t\})$ is the remapping from BEV to front. Since DBEV is parameterized by the pitch and roll, which is related to the variance of IMU data, with the help of DUKF, the gradient can be back-propagated to the variance.

## 2.3 Differentiable Phase Correlation

Given two images $I_{t-1}^{bev}$ and $I_t^{bev}$ with pose transformations, a variation of the differentiable phase correlation [18] is utilized to estimate the the overall relative pose $\xi_i$ between $I_{t-1}^{bev}$ and $I_t^{bev}$. The rotation and scale are calculated by:

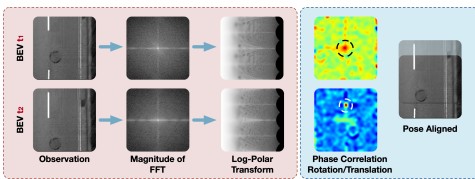

Figure 3: The **visual demonstration** DPC.

$$p(\xi_i^{\theta,s}) = \mathfrak{C}(\mathfrak{L}(\mathfrak{F}(I_t^{bev})), \mathfrak{L}(\mathfrak{F}(I_{t-1}^{bev}))) \qquad (19)$$

$$(\theta_t, s_t) = \mathbb{E}(p(\xi_i^{\theta,s})), \qquad (20)$$

where $\xi_t^{\theta,s}$ is the rotation and scale part of $\xi_i$ between $I_{t-1}^{bev}$ and $I_t^{bev}$, $\mathfrak{F}$ is the discrete Fourier Transform, $\mathfrak{L}$ is the log-polar transform Log-Polar Transform), and $\mathfrak{C}$ is the phase correlation solver. Fourier Transformation $\mathfrak{F}$ transforms images into the Fourier frequency domain of which the magnitude has the property of translational insensitivity, therefore the rotation and scale are decoupled with displacements and are represented in the magnitude (Fig. 3 Magnitude of FFT). Log-polar transformation $\mathfrak{L}$ transforms Cartesian coordinates into log-polar coordinates so that such rotation and scale in the magnitude of Fourier domain are remapped into displacement in the new coordinates, making it solvable with the afterward phase correlation solver (Fig. 3. Phase correlation solver $\mathfrak{C}$ outputs a heatmap indicating the displacements of the two log-polar images, which eventually stands for the rotation $\theta_t$ and scale $s_t$ of the two input images $I_{t-1}^{bev}$ and $I_t^{bev}$ (Fig. 3 Phase Correlation of Rotation). Since we assume that the height of the robot does not change drastically along the trajectory, the scale $s_t$ is fixed as 1. To make the solver differentiable, we use expectation $\mathbb{E}(\cdot)$ as the estimation of $\because$.

Then $I_t^{bev}$ is rotated and scaled referring to $\xi_t^{\theta,s}$ with the result of $\bar{I}_t^{bev}$. In the same manner, translations $\xi_t^{\mathbf{t}}$ between $\bar{I}_t^{bev}$ and $I_{t-1}^{bev}$ is calculated with $p(\xi_t^{\mathbf{t}}) = \mathfrak{C}(\bar{I}_t^{bev}, I_{t-1}^{bev})$ and $\mathbf{t}_t = \mathbb{E}(p(\xi_t^{\mathbf{t}}))$. With the relative rotation $\theta_t$, scale $s_t$, and translation $\mathbf{t}_t$ between $I_{t-1}^{bev}$ and $I_t^{bev}$, and by consulting with the fixed $Distance\ per\ Pixel$ (DPP), we finally arrive at the $\mathbb{SIM}(2)$ pose between frame $t-1$ and $t$ of vehicle:

$$\boldsymbol{S}_t = [\theta_t, \quad s_t, \quad DPP \cdot \mathbf{t}_t]. \qquad (21)$$

Note that the gradient in the part can be back-propagated all the way back to the variance of the IMU data since the phase correlation is differentiable.

## 2.4 Loss Design, Back-Propagation, and Training

With DUKF, DBEV and DPC, we define a one-step VIO as a function:

$$\{\theta_t, s_t, DPP \cdot \mathbf{t}_t, \alpha_t, \beta_t\} = \mathcal{VIO}_{\boldsymbol{O},\boldsymbol{Q}}(I_{t-1}, I_t, \omega, acc), \qquad (22)$$

where $\mathcal{VIO}_{\boldsymbol{O},\boldsymbol{Q}}$ is fully differentiable and interpretable with 4 trainable parameters in $\boldsymbol{O}_t$ and $\boldsymbol{Q}_t$. $I$, $\omega$, and $a$ are the front images, gyroscope data, and accelerometer data. Since the BEVO is fully differentiable, it is trained end to end and supervised only with the ground truth of the odometry

between frames. Such supervision is applied to optimize the covariance matrices of noise in both the motion model and the measurement model of DUKF.

Elaborately, we wish to supervise the distribution of the poses $\xi_t^{\theta,s}$ and $\xi_t^{\mathbf{t}}$ and therefore, we calculate the Kullback–Leibler Divergence loss $KLD$ between $\{p(\xi_t^{\theta,s}), p(\xi_t^{\mathbf{t}})\}$ and the ground truth:

$$\mathcal{L} = KLD(p(\xi_t^{\theta,s}), \mathbf{1}_{\theta^*}) + KLD(p(\xi_t^{\mathbf{t}}), \mathbf{1}_{\mathbf{t}^*}), \tag{23}$$

where $\mathbf{1}_{\theta^*}$ and $\mathbf{1}_{\mathbf{t}^*}$ are the Gaussian Blurred one-peak distributions centered at the ground truth $\theta^*$ and $\mathbf{t}^*$ respectively.

**Training Details:** all trainings are conducted on a GPU server with NVIDIA RTX 3090. For the initialization, we randomize all initial values of $O$ and $Q$ within the range of $[0, 1]$ uniformly. Since $O$ and $Q$ are $2\times2$ diagonal matrices, we consider the learning process of the spherical covariances of them following [1][21] so that the number of trainable parameters will be reduced to 4. We supervise the model only with 3 DoF ground truth (GT) data of the same frequency as the images, i.e. the high frequency GT of the IMU is not needed. We set the BEV size to $100 \times 100$ as a trade-off between speed and accuracy. The inference time is $21ms$ and takes up about $500MB$ of the resources. The pseudocode for training BEVO will be given in Appendix I.

**Plug into Map-Based Localization System:** in addition to the odometry, we also explore the possibility of integrating BEVO with fully differentiable localization to achieve a drift-free system. We show results as demonstration of the application along with the pseudocode shown in Appendix II.

## 3 Experiment

We conduct two main experiments: i) odometry estimation on the KITTI [22] odometry dataset on which the experiments on ablation study, testing, generalization are conducted, and ii) heterogeneous localization on CARLA [23] and the AeroGround Dataset [24].

**Dataset and Corresponding Experiments:**

*KITTI & CUD:* We evaluate BEVO on the KITTI odometry dataset using sequences 00-08 for training and 09-10 for the test following the same train/validate/test split as [25][26]. In addition to the widely adopted train/validate split where $20\%$ samples are randomly sampled from the training set for validation, we also train our model in the first three quarters of each sequence of 00-08, and evaluate the performance in the last quarter. This train/validate split is more realistic even though it might present a worse result. Moreover, we further investigate the generalization ability by training BEVO with $20\%$ data of each sequence. Note that sequence 03 cannot be used for training or evaluation in this study since the raw data where we grab the IMU data is missing.

*CARLA & AG:* We further verify the plausibility of applying the BEVO as a front-end plugin for end-to-end heterogeneous localization following the routine in [27]. Such experiments are conducted in CARLA and on the real-world AeroGround Dataset. The setups are elaborated in Appendix III.

**Evaluation Matrices:** We evaluate our trajectories using $t_{rel}$ and $r_{rel}$ from the standard KITTI toolkit as in [26]. In our case, we do not estimate the elevation due to the assumption that the vehicle will not leave the ground. However, we still record the initial elevation and keep it constant through the journey for comparison, which might **degrade** our performance in $t_{rel}$ to some extent.

**Baselines:** For the odometry experiment, we compare to both traditional method VINS [1] and SOTA depth dependent learning-based methods DEEPVIO [5], SELFVIO [28], LI ET AL. [26], GEONET [29], and depth independent learning-based method TARTANVO [30]. For the localization experiments, we compare the performance with VINS-MONO [1] and ORB-M [21].

### 3.1 Odometry Estimation

We now compare the odometry estimation performance of the proposed BEVO to the state-of-the-art methods in two training settings as introduced in **Dataset**: i) following the train/validate splits in [25][26] (**Ours**[1]), and ii) split each sequence by quarters, take the first three quarters of each sequence to train and the last quarter to validate (**Ours**[2]). In addition to these two settings, we further train BEVO using the first $20\%$ of the training data of each sequence in **Ours**[2] to evaluate its performance on insufficient data (**Ours**[3]).

TABLE 1: **Quantitative results** of odometry estimation on KITTI dataset from sequence 00 to 10. Red indicates the best performance and Blue indicates the second best.

| Seq | frames | DeepVIO $t_{rel}$ | $r_{rel}$ | TartanVO $t_{rel}$ | $r_{rel}$ | Li et al. $t_{rel}$ | $r_{rel}$ | GeoNet $t_{rel}$ | $r_{rel}$ | SelfVIO $t_{rel}$ | $r_{rel}$ | VINS $t_{rel}$ | $r_{rel}$ | ORB-M $t_{rel}$ | $r_{rel}$ | Ours[1] $t_{rel}$ | $r_{rel}$ | Ours[2] $t_{rel}$ | $r_{rel}$ | Ours[3] $t_{rel}$ | $r_{rel}$ |
|---|---|---|---|---|---|---|---|---|---|---|---|---|---|---|---|---|---|---|---|---|---|
| 00 | 4541 | 11.62 | 2.45 | \ | \ | 14.21 | 5.93 | 44.08 | 14.89 | 1.24 | 0.45 | 18.83 | 2.49 | 25.29 | 7.73 | 3.26 | 1.44 | 1.55 | 0.83 | 2.01 | 1.02 |
| 01 | 1101 | \ | \ | \ | \ | 21.36 | 4.62 | 43.21 | 8.42 | \ | \ | \ | \ | \ | \ | 5.62 | 3.95 | 1.17 | 1.66 | 1.76 | 2.13 |
| 02 | 4661 | 4.52 | 1.44 | \ | \ | 16.21 | 2.60 | 73.59 | 12.53 | 0.80 | 0.25 | 21.03 | 2.61 | 26.30 | 3.10 | 4.56 | 1.32 | 1.02 | 0.88 | 1.79 | 1.55 |
| 04 | 271 | \ | \ | \ | \ | 9.08 | 4.41 | 17.91 | 9.95 | \ | \ | \ | \ | \ | \ | 3.79 | 1.05 | 2.04 | 0.92 | 3.12 | 1.15 |
| 05 | 2761 | 2.86 | 2.32 | \ | \ | 24.82 | 6.33 | 32.47 | 13.12 | 0.89 | 0.63 | 21.90 | 2.72 | 26.01 | 10.62 | 2.05 | 1.57 | 0.72 | 1.01 | 1.55 | 1.80 |
| 06 | 1101 | \ | \ | 4.72 | 2.95 | 9.77 | 3.58 | 40.28 | 16.68 | \ | \ | \ | \ | \ | \ | 3.86 | 2.08 | 1.07 | 1.56 | 2.77 | 1.99 |
| 07 | 1101 | 2.71 | 1.66 | 4.32 | 3.41 | 12.85 | 2.30 | 37.13 | 17.20 | 0.91 | 0.49 | 15.39 | 2.42 | 24.53 | 10.83 | 3.58 | 1.47 | 0.88 | 1.35 | 1.15 | 1.79 |
| 08 | 4701 | 2.13 | 1.02 | \ | \ | 27.10 | 7.81 | 11.45 | 62.45 | 1.09 | 0.36 | 32.66 | 3.09 | 32.40 | 12.13 | 1.94 | 1.04 | 0.93 | 0.92 | 1.29 | 1.00 |
| 09 | 1591 | 1.38 | 1.12 | 6.0 | 3.11 | 15.21 | 5.28 | 13.02 | 67.06 | 1.95 | 1.15 | 41.47 | 2.41 | 45.52 | 3.10 | 1.22 | 1.05 | 1.29 | 1.12 | 1.32 | 1.31 |
| 10 | 1201 | 0.85 | 1.03 | 6.89 | 2.73 | 25.63 | 7.69 | 58.52 | 23.02 | 1.81 | 1.30 | 20.35 | 2.73 | 6.39 | 3.20 | 1.01 | 1.01 | 1.14 | 1.03 | 1.17 | 1.30 |
| Ave 00-10 | | 3.73 | 1.57 | 5.48 | 3.05 | 16.02 | 4.60 | 33.78 | 22.30 | 1.24 | 0.67 | 21.61 | 2.64 | 26.65 | 4.67 | 2.80 | 1.45 | 1.07 | 1.02 | 1.63 | 1.36 |
| Ave 09-10 | | 1.12 | 1.08 | 6.74 | 2.92 | 20.42 | 6.49 | 35.77 | 45.04 | 1.88 | 1.23 | 30.91 | 2.57 | 25.96 | 3.15 | 1.12 | 1.03 | 1.22 | 1.07 | 1.24 | 1.31 |

Fig.4 and Table 1 show the qualitative and quantitative comparisons, respectively. Table 1 shows that when trained in the same settings as other baselines, **Ours**[2] outperforms most of the baselines in validation results in sequence 00 to 08. However, regarding the unseen testing sequences, our **Ours**[1] beats all other baseline in both $t_{rel}$ and $r_{rel}$ except for the $t_{rel}$ of DEEP-VIO in sequence 10. This may be explained by the fact that DEEPVIO utilizes stereo cameras instead of monocular camera considered in our method. The result of the testing sequence shows that our method is better at generalization compared to other learning-based methods. Fig. 4 also shows that, even though pitch and roll are not directly supervised, but with the low-frequency image-level supervision in 3 DoF *only*, they are still accurate. This proves the effectiveness of the end to end training and that the interpretability provides inductive bias, which accounts for a good generalization.

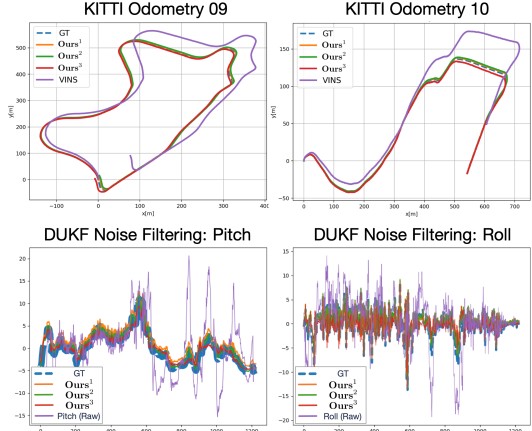

Figure 4: The **qualitative results** of **Top:** the evaluation on KITTI sequence 09, and 10, **Bottom:** Noise filtering of the pitch and roll angle through sequence 10. Visual results on other sequences can be found in Appendix IV.

It is worth addressing that without employing any nonlinear optimization such as sliding window, our method is surprisingly comparable with others on sequences 00-08, and is even slightly better in testing on sequences 09 and 10. The result of **Ours**[3] suggests that, in contrast to learning-based approaches reliant on a large number of training images, our method is more data-efficient thanks to the fully interpretable model.

### 3.2 Plugin for Heterogeneous Localization

In this part, we show the possibility of enabling an end-to-end localization with BEVO as the odometry. Elaborations on the implementation details of the fully differentiable drift-free localization can be found in the supplementary material. By combining BEVO with the heterogeneous localization (denoted as BEVO+), the whole pipeline is fully differentiable, thus allowing for more accurate pose estimation. We compare the localization BEVO+ with BEVO itself as well as the VINS-MONO (denoted as VINS in the table). Results in Fig. 5 and Table 2 prove that the BEVO is able to achieve good performance even when it acts as a plugin in an end-to-end trainable localization framework. It indicates a promising future for upcoming end-to-end autonomous driving, e.g., auto-wheelchair.

### 3.3 Ablation Study

Several experiments conducted on KITTI are designed for the ablation study to reassure the important role a DUKF plays in the BEVO as well as the role trainable covariance matrices play in DUKF. First, we replace the learned covariance matrix $O_t$ by one that is empirically set in [20]. The quantitative result in Table. 3 shows that the accuracy drops drastically when us-

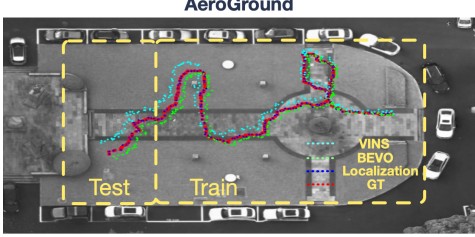
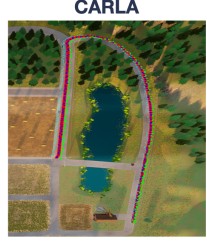

**AeroGround**  **CARLA**

Figure 5: The **visual demonstration** of localizing a front camera's BEV of a ground moving robot in drone's map. More Qualitative demonstration are shown in the Appendix V.

TABLE 2: **Quantitative results** of localization. Note: in CARLA, we localize the vehicle in different weathers against a sunny satellite map. (S-sunny, F-foggy, N-night)

| Dataset | BEVO | | VINS | | **BEVO+** | |
|---|---|---|---|---|---|---|
| | $t_{rel}$ | $r_{rel}$ | $t_{rel}$ | $r_{rel}$ | $t_{rel}$ | $r_{rel}$ |
| CARLA-S | 1.05 | 1.82 | 2.06 | 2.58 | 0.21 | 0.33 |
| CARLA-F | 1.11 | 1.38 | 2.92 | 2.14 | 0.24 | 0.31 |
| CARLA-N | 0.98 | 1.03 | 1.82 | 1.47 | 0.19 | 0.28 |
| AG | 1.39 | 1.72 | 2.55 | 2.41 | 0.30 | 0.46 |

ing a manually tuned matrix, suggesting that the empirical parameter tuning leads to inaccurate pitch and roll estimation, and subsequently deteriorates the BEV generation and odometry estimation. Next, we restore $O_t$ with the learned matrix and replace $Q_t$ with an empirical covariance matrix as [20], and note it as "w/o $Q_t$". The quantitative result also verifies that trainable covariance matrices for noise help to improve the performance of UKF. However, even though the performances are degraded with empirical parameters, they are still competitive due to the dense matching of DPC who seeks global optima. We further study the importance of DUKF on the BEVO by eliminating the whole DUKF and feeding the raw pitch and roll (directly calculated from the accelerometer) to the BEV projection. We note it as "w/o DUKF".

The quantitative result shown in Table. 3 proves that when the filter is gone, the BEVO will encounter serious tracking failure in bumpy roads. One more ablation study is conducted by replacing the output pitch and roll of DUKF with the ground truth, denoted as "w/ GT". The result that BEVO is comparable with "w/ GT" indicates that by learning the covariance matrices, the error brought by DUKF is too small to influence the odometry than the error by DPC, showing the effectiveness of the training.

TABLE 3: **Quantitative results** of ablation study conducted on KITTI. All of the methods are trained in sequence 00-08 following the train spit in [26] and tested in sequence 09-10.

| | w/o $O_t$ | | w/o $Q_t$ | | w/o DUKF | | **BEVO** | | w/ GT | |
|---|---|---|---|---|---|---|---|---|---|---|
| Seq | $t_{rel}$ | $r_{rel}$ | $t_{rel}$ | $r_{rel}$ | $t_{rel}$ | $r_{rel}$ | $t_{rel}$ | $r_{rel}$ | $t_{rel}$ | $r_{rel}$ |
| 09 | 4.82 | 4.73 | 5.90 | 5.11 | 22.05 | 71.58 | 1.29 | 1.12 | 1.05 | 1.04 |
| 10 | 5.66 | 4.87 | 6.26 | 5.62 | 25.64 | 68.54 | 1.14 | 1.03 | 0.96 | 1.01 |

## 3.4 Limitation

Though being competitive in odometry estimation for most of the ground mobile robots, BEVO, limited by the assumption of constant elevation, is yet hardly applicable to robots that move with large elevation changes, e.g. drones. However, since the phase correlation in Section 2.3 is able to estimate the scale changes along with the rotation (yaw) between images, it should potentially serve as a way to calculate the relative elevation changes. We will address the above limitations in future works and hopefully make BEVO compatible in all scenarios.

## 4 Conclusion

We present a fully differentiable, interpretable and lightweight model for monocular VIO, namely, *BEVO*. It comprises a UKF to denoise pitch and roll via trainable covariance matrices, a BEV projection, and phase correlation to estimate the $\mathbb{SIM}(2)$ pose on the BEV image. The differentiability allows us to train the covariance matrices end-to-end, thus avoiding empirical parameter tuning. In various experiments, *BEVO* presents a competitive performance in accuracy and generalization. We believe our interpretable, simple approach provides an alternative perspective of traditional methods and should be considered as a baseline for future works on learning-based monocular VIO.

**Acknowledgments**

This work was supported in part by the National Key R&D Program of China (Grant No.2020YFB1313300), Zhejiang Provincial Natural Science Foundation of China (LD22E050007), and Science and Technology on Space Intelligent Control Laboratory (2021-JCJQ-LB-010-13). We also thank reviewers who gave useful comments.

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
