# OpenReview forum: "Learning Interpretable BEV Based VIO without Deep Neural Networks"
_robot-learning.org/CoRL/2022/Conference — CoRL 2022 Poster_

### Official Review · Reviewer_p14S · 2022-07-21

**Originality:** Excellent
**Technical Quality:** Excellent
**Clarity Of Presentation:** Excellent
**Impact:** 4

**Recommendation:**

Strong Accept: I recommend accepting the paper and will argue for my recommendation even if other reviewers hold a different opinion.

**Summary:**

This work introduced a novel visual-inertial odometry system, which is learning-based but interpretable without deep neural networks. Specifically, the IMU data is used by a differentiable UKF to estimate the roll and pitch, which are subsequently used to project images to bird-eye view (BEV). The BEV images are transformed to the frequency domain to estimate SIM(2) camera motion. The leaning part comes from the UKF covariance; the authors maintain the following BEV projection and motion estimation differentiable so that the odometry error could be back-propagated to the UKF covariance, which makes it learnable and the estimated roll/pitch optimal. The evaluation is mainly based on the standard KITTI dataset. The proposed method achieves great results.

**Issues:**

- Line 121: ‘within the red dashed box’, do you mean ‘within the yellow dashed box’?
- In Section 2.3, the ‘scale’ term is slightly confusing. Since the proposed method is estimating SIM(2) motion, the reviewer assumes the ‘scale’ refers to the odometry scale instead of image scaling. If so, what is the reason to fix the scale to 1 (line 150)? fixing scale to 1 makes SIM(2) to SE(2), right?
- How do the authors handle the IMU bias terms? are they not modeled in the UKF?


**Quality Of The Limitations Section:**

Limitations are addressed clearly

**Reviewer Expertise:**

4: The reviewer is confident but not absolutely certain that the evaluation is correct

**Robotics Focus:**

Sufficient demonstration on hardware

**Strengths And Weaknesses:**

Strength
- The paper is well-written, and the methodology is clearly stated and technically sound. It is a really solid work with significant novelties and contributions. It is refreshing to review a learning-based VIO work that is not based on deep networks.

Weakness
- The unchanged height assumption for driving scenarios seems fine; however, the reviewer is slightly concerned about the sufficient distance assumption: what is the minimal distance? is it also related to the camera mounting height and angle? if there are objects in that yellow box, how will it distort the BEV images and decrease the VIO accuracy?
- The evaluation is mainly based on the KITTI dataset, focusing on sequences 09-10 for testing. The validation would be more persuasive if the authors include more VIO testing.


**Summary Of Recommendation:**

It is a really solid work with novel method.

---

> ### Author Response · Authors · 2022-08-21
> **Response to Reviewer p14S (5/5)**
>
> > **How do the authors handle the IMU bias terms? are they not modeled in the UKF?**
>
> Thank you for your question! It is true we did not model the IMU bias term in the UKF module since we found that the filtered result is acceptable and doesn't affect the verification of the hypothesis that a fully interpretable and learnable model can achieve better generalization. We really appreciate your suggestions and will model the IMU bias term in our future filter module to make the estimation more accurate. We have also included this future work in the *Limitation*.
>
>
>
> As a conclusion, following the advice, we carefully revised the manuscript with:
>
> - We added an extra explanation about the sufficient distance to discuss the minimal distance needed and how the distance will be affected in *Section 2.2.*
> - We added more experiments on KAIST's Complex Urban Dataset (CUD) in the appendix to better verify the cross-scene generalization performance of our algorithm.
> - We fixed our mistake about the dashed box color in Line 121 which should be "yellow".
> - We improved our statement about the "scale" issue of Differentiable Phase Correlation in *Abstract* and in Section 2.3 to make it more clear and more specific.

---

> > ### Comment · Reviewer_p14S · 2022-08-27
> > **Thank you for your explanation and clarification.**
> >
> > Thank you for your explanation and clarification.

---

> > > ### Author Response · Authors · 2022-08-27
> > > **Many Thanks**
> > >
> > > Thank you again for the wonderful advice!

---

> ### Author Response · Authors · 2022-08-21
> **Response to Reviewer p14S (4/5)**
>
> > **In Section 2.3, the ‘scale’ term is slightly confusing. Since the proposed method is estimating SIM(2) motion, the reviewer assumes the ‘scale’ refers to the odometry scale instead of image scaling. If so, what is the reason to fix the scale to 1 (line 150)? fixing scale to 1 makes SIM(2) to SE(2), right?**
>
> Thank you for this question. We apologize that our statement may cause your misunderstanding.  We refer to the word ‘scale’ as the scale between two BEV images. Estimating poses between two BEV images is also a SIM(2) problem. However, since we estimate local planar motion, the height of the sensor to the ground is assumed to be constant, therefore the scale between two BEV images is fixed as 1. That means in the phase correlation, we are estimating the SE(2) motion of the robot: planar translation, and yaw rotation. Combining with the IMU from the DUKF that estimates the roll and pitch angle, we finally come up with a 5DoF estimation. This part was addressed in the *Methodology*, and we have revised it to make it clearer.

---

> ### Author Response · Authors · 2022-08-21
> **Response to Reviewer p14S (3/5)**
>
> > **Line 121: ‘within the red dashed box’, do you mean ‘within the yellow dashed box’?**
>
> Thank you for pointing out our mistake in paper writing. We apologized for this mistake and have corrected it in the revision.

---

> ### Author Response · Authors · 2022-08-21
> **Response to Reviewer p14S (2/5)**
>
> > **The evaluation is mainly based on the KITTI dataset, focusing on sequences 09-10 for testing. The validation would be more persuasive if the authors include more VIO testing.**
>
> Thank you for mentioning this! In the original manuscript, we experimented on the Carla simulation and real-world AeroGround dataset for heterogeneous end-to-end localization using BEVO as a front-end plug-in, which proved the efficiency and accuracy of our VIO algorithm.
>
> Moreover, as you have suggested, we have also carried out additional experiments on KAIST's Complex urban dataset (CUD) which is illustrated in the revised appendix. We train BEVO in scene ''Urban30'',''Urban31'',''Urban32'' and test the performance in scene "Urban33", ''Urban34'', and ''Urban35''. Note these sequences are collected by the same device, but from different locations. Due to the tight timeline, we did not train other baselines in CUD to compare but only demonstrated the performance of BEVO, however, we will further compare with other baselines in the camera-ready version.

---

> ### Author Response · Authors · 2022-08-21
> **Response to Reviewer p14S (1/5)**
>
> Thank you for your valuable feedback. We address your concerns below, and have made revisions to the paper (highlighted in orange) based on your comments, which we believe has improved the paper. Please let us know if you have any remaining concerns or questions!
>
> > **The unchanged height assumption for driving scenarios seems fine; however, the reviewer is slightly concerned about the sufficient distance assumption: what is the minimal distance? is it also related to the camera mounting height and angle? if there are objects in that yellow box, how will it distort the BEV images and decrease the VIO accuracy?**
>
> Thank you for your question! We empirically set the distance to 10 meters, similar to existing BEV based methods [1][2]. We expect reducing this distance may worsen the performance as our differentiable phase correlation module relies on image textures. The distance is indeed relevant to the camera mounting height and angle to assure sufficient image textures can be kept in the yellow box.
>
> At the methodology level, DPC does not assume that all objects in the BEV image are static. It will work when the cost for aligning static objects is smaller than that for dynamic objects. It also means that DPC can not treat BEV images with large amount of dynamic objects. However, it is possible to eliminate dynamic objects by leveraging semantic information [1]. While this work focuses more on the learning of interpretable models, we did not explore deeper in the dynamics, which we believe will be included in future works. Thank you again for this question and we have included this in the *Limitation*.
>
> *[1]A Light-Weight Semantic Map for Visual Localization towards Autonomous Driving, Qin et al.*
>
> *[2]Avp-slam: Semantic visual mapping and localization for autonomous vehicles in the parking lot, Qin et al.*

---

### Official Review · Reviewer_3Som · 2022-07-25

**Originality:** Good
**Technical Quality:** Fair
**Clarity Of Presentation:** Good
**Impact:** 2

**Recommendation:**

Strong Reject: I recommend rejecting the paper and will argue for my recommendation even if other reviewers hold a different opinion.

**Summary:**

This paper proposed an interpretable architecture for VIO, which admits differentiable training. Experimental results on two domains demonstrate that the proposed approach is effective and competitive against recent deep learning-based architectures.

**Issues:**

None.

**Quality Of The Limitations Section:**

Limitations are addressed clearly

**Reviewer Expertise:**

2: The reviewer is willing to defend the evaluation, but it is quite likely that the reviewer did not understand central parts of the paper

**Robotics Focus:**

Highly relevant to robotics but no hardware experiments

**Strengths And Weaknesses:**

Strengths:

The architecture is novel, as far as I can tell. It also achieves very high performance.

Weaknesses:

Let me preface my comment by saying that I am not an expert on VIO, but come more from the general model interpretability side. With that said, I am quite disappointed about the lack of empirical evaluation of the model interpretability. While the model achieves comparable performance as deep models, which is refreshing to see as many interpretable models struggle to match the performance of their non-interpretable counterparts, it is not clear in what ways the proposed model is exactly interpretable.

Demonstrating interpretability requires validating that people, developers and/or users, can gain concrete understanding of the model. Thus, at the very least, there should be more analysis on the model itself, besides that on model performance. Ideally, it would also show some specific aspects corresponding to the model understanding. For example, maybe the developer can identify some issues of the model so that they can re-train a better one, or that a domain expert can understand the weakness of it, so that they can choose when not to trust the model prediction. Without any of these demonstrations, I am not convinced that this model is any more interpretable than a standard neural network.

**Summary Of Recommendation:**

Due to the lack of experiments and analyses on the interpretablity front, I could not recommend this paper for publication. I believe that there could be two avenues forward:

1. Remove all claims of interpretability, so that this paper's contribution is limited to a novel architecture for VIO that matches SoTA performance. Or better,

2. Conduct experiments to validate interpretability, and position this model as an interpretable and high-performing one.

---

> ### Author Response · Authors · 2022-08-21
> **Response to Reviewer 3Som (2/2)**
>
> > * **Ideally, it would also show some specific aspects corresponding to the model understanding. For example, maybe the developer can identify some issues of the model so that they can re-train a better one, or that a domain expert can understand the weakness of it, so that they can choose when not to trust the model prediction.**
> >
> > * **Or better, ii) Conduct experiments to validate interpretability, and position this model as an interpretable and high-performing one.**
>
> Thank you for pointing this out. If the reviewer can kindly refer to the ablation study in the manuscript, we have actually conducted experiments eliminating the learnable parameters in the method to restore the method to a mechanistic one. These parameters have specific definitions in the mechanistic model and are given by data sheet, calibration, or expert experiences in traditional methods, which means the users do have a concrete understanding of the parameters. In the ablation study, experts designed a set of parameters based on [2] to replace the learned one, and the method still works, but could not find a better one to achieve competitive performance to the data-driven parameters. Also, taking the poor performance of traditional methods VINS and ORB-M in KITTI for example, the low IMU frequency and the uncritical sync of the IMU make it hard for experts to design a perfect parameter.
>
> * Therefore we argue that by combining interpretable mechanistic models with end-to-end learning of a few parameters, the inherent interpretability is not diminished, but even will thrive with more suitable parameters. This can be justified by the ablation study.
>
> *[2]The Unscented Kalman Filter, E.A et al.*

---

> > ### Author Response · Authors · 2022-08-27
> > **Dear reviewer 3Som**
> >
> >
> > Thank you again for your review. We hope that our rebuttal could address your questions and concerns. As the discussion phase is nearing its end, we would be grateful to hear your feedback and wondered if you might still have any concerns we could address.
> > Thank you for your time.

---

> ### Author Response · Authors · 2022-08-21
> **Response to Reviewer 3Som (1/2)**
>
> Thank you for your valuable feedback. We address your concerns below, and have made revisions to the paper (highlighted in orange) based on your comments, which we believe has improved the paper. Please let us know if you have any remaining concerns or questions!
>
> > * **While the model achieves comparable performance as deep models, which is refreshing to see as many interpretable models struggle to match the performance of their non-interpretable counterparts, it is not clear in what ways the proposed model is exactly interpretable.**
> >
> > * **Demonstrating interpretability requires validating that people, developers and/or users, can gain concrete understanding of the model. Thus, at the very least, there should be more analysis on the model itself, besides that on model performance.**
>
> Firstly, we apologize if we did not make clear how our method is interpretable. We would like to clarify the interpretability of the method from our point of view. Citing from [1]: *"Interpretable ML focuses on designing models that are inherently interpretable; whereas explainable ML tries to provide post hoc explanations for existing black box models"*, we would like to reclaim that we are not trying to explain a black box but to design a model that is inherently interpretable. If results are learned using black-box components like FCN, it is uninterpretable, let alone design non-trainable components to replace it. However, our model does not contain such uninterpretable components, and the four learnable parameters have a user-understandable meaning: the covariances of IMU measurements. Users can  empirically set these four parameters to replace the learned one, like what has been tested in the ablation study, the model will still work, just not as good as the data-driven trained parameters.
>
> To be concrete on how the method is interpretable, if we deprecate all the learnable parts of the method, the backbone of it will emerge as a purely mechanistic model with a parsing solution:
>
> * Unscented Kalman Filter (UKF) $\rightarrow$ Bird-eye View Projection (BEV) $\rightarrow$ Phase Correlation (PC)
>
> Essentially, mechanistic models are generally dynamic models based on a description of processes that we believe are happening in a system, and are usually given by interpretable equations. Such equations are given in the manuscript as well as their pseudo-code in the appendix. By saying this, we argue that mechanistic models are interpretable, and thus when we differentiate them to build an end-to-end VIO, it is inherently interpretable. However, the mechanistic model requires handcrafted tuning for the covariances of the IMU measurements. Now look back to BEVO with learnable parts, the only difference compared to the above interpretable model is whether these two covariance matrices are learned from data or not. Moreover, the definition of these two matrices is clear: the covariances of noise in angular rate and acceleration measured by the IMU. By clarifying this, we believe that the developers and/or users, should gain a concrete understanding of the model, since:
>
> * the method is derived from the interpretable mechanistic model, of which the equations are given in the manuscript and the appendix,
> * the parameters we trained have a clear definition: the covariances of IMU measurements.
>
> *[1]Stop Explaining Black Box Machine Learning Models for High Stakes Decisions and Use Interpretable Models Instead, Rudin et al.*

---

### Official Review · Reviewer_vD2G · 2022-07-30

**Originality:** Good
**Technical Quality:** Good
**Clarity Of Presentation:** Good
**Impact:** 2

**Recommendation:**

Weak Reject: I recommend rejecting the paper, but will not argue for my recommendation if the majority of other reviewers have a different opinion.

**Summary:**

This paper proposes a visual inertial odometry system based on kalman filtering. The main idea of this work is to learn the noise filtering parameters by making the filter differentiable. While the idea of learning filters is not new (See [1]), the application to the problem of VIO from a bird-eye perspective is new to me.  The approach is evaluated against a set of baselines on the KITTI dataset, where it obtains comparable or better results.


[1] Krishnan et al,, Deep Kalman Filters

**Issues:**

I would be happy to see a more aligned introduction and experimental section in the revision. Also, I would like to see a discussion of previous methods and a clearer answer to the question: "When are algorithmic priors a good idea?"

**Quality Of The Limitations Section:**

Limitations are not well addressed

**Reviewer Expertise:**

3: The reviewer is fairly confident that the evaluation is correct

**Robotics Focus:**

Relevant but unlikely to deploy to hardware in near future

**Strengths And Weaknesses:**

The main strength of the paper is the nice balance between end-to-end learning and interoperability. Learning only a few parameters end-to-end makes the system easy to debug and investigate. It is to be said, however, that similar ideas have been long investigated in the community (See [1],  [2] or many others), but none really took off. This could be because learning so few parameters makes the algorithm strongly reliant on algorithm priors, which might be inappropriate in several real-world applications.
A second interesting takeaway from this work is that traditional VIO methods, like VINS, perform terribly on KITTI. However, I find this somehow difficult to believe since, in the KITTI benchmark, most monocular systems are traditional VIO methods. Is this setup somehow different? What makes it challenging for them in the proposed evaluation?
Finally, I find the idea of passing through the Bird-eye view quite interesting. I would have expected to be a suboptimal choice given the presence of other vehicles and humans, whose upper part is generally occluded. However, this does not seem to be a problem for the proposed system, which I find quite interesting.

The main weakness of the approach is, in my opinion, the evaluation setup and results. As mentioned before, the terrible performance of traditional VIO methods is difficult to interpret. In addition, I feel that the main claim of this paper (rely more on algorithmic priors than parameters) is not sufficiently justified. It would be interesting to see results on more challenging datasets, possibly in 3D, where the assumption on the motion model would not always hold. If the paper only wishes to specialize in AV scenarios, then such choices must be made clear from the beginning (abstract/intro) and claims weakened. The limitation section mentions that the approach would be applicable for all ground robots, but there is not enough evidence to support this claim. Overall, the main question I feel should be answered is: when is desirable to enforce algorithmic priors instead of parameters?
Another weakness is the fact that previous work on deep state estimators (like [1],[2], and others) has not been sufficiently covered and discussed. In what does the approach differ? Why is it better?
Finally, a (minor) limitation is that the approach is said to be lightweight, but computation time is not compared to existing methods.

[1]  Deep Kalman Filters, Krishnan et al.
[2] Differentiable Particle Filters: End-to-End Learning with Algorithmic Priors, Jonschkowski et al.


**Summary Of Recommendation:**

My recommendation is based on the disparity between the claims and the provided results. In addition, the paper does not make a strong case on why this approach would be relevant in general ground robotic applications (AVs also have GPS, so why not use it?).

---

> ### Author Response · Authors · 2022-08-21
> **Response to Reviewer vD2G (7/7)**
>
> > **I would be happy to see a more aligned introduction and experimental section in the revision.**
>
> Thank you for raising this!
>
> Considering the explanation above, we conclude the revision of the manuscript as follows:
>
> * We have corrected the *Introduction* and discussed the suggested related works.
> * We have included the analysis for the poor performance of VINS as well as the results for another traditional ORB-M in the *Experiments* in the revision.
> * We have made clear the specialization in AV scenarios in *Abstract* and *Introduction* and downtoned the claim so that the it is more aligned with the experimental results.
> * We added more experiments on KAIST's Complex Urban Dataset in the *appendix* to justify the cross-scene generalization ability.
> * We expanded the discussion of the relationships between the BEV size and the performances (runtime and accuracy) in the *appendix*.
> * We have included the relations between BEVO and  general ground robotic applications in the *Introduction*.

---

> ### Author Response · Authors · 2022-08-21
> **Response to Reviewer vD2G (6/7)**
>
> > **In addition, the paper does not make a strong case on why this approach would be relevant in general ground robotic applications (AVs also have GPS, so why not use it?).**
>
> Thank you for this question. There indeed exist many odometries that merge GPS into the estimation that will lift the precision. However, they tend to be hindered in GPS-denied places, such as underground parking lot and tunnels. Therefore, many odometries aim at onboard sensors that do not require outer communications and they are validated in KITTI without GPS information. This reflects that the AV problem we are dealing with is an important application in general ground robotics.
>
> We really appreciate this insightful advice that shows us an alternative way forward to improve the method by merging the GPS into the filter when it is available. We have included the above statement in the *Introduction* so that the application of BEVO could be clearer.

---

> ### Author Response · Authors · 2022-08-21
> **Response to Reviewer vD2G (5/7)**
>
> > **Finally, a (minor) limitation is that the approach is said to be lightweight, but computation time is not compared to existing methods.**
>
> Thanks for this advice. In all of our experiments, the inference time for one motion is 21ms, and the memory usage is 500MB. This is obtained using BEV images with a resolution of 100x100. Intuitively, the size of the BEV images affects runtime, memory usage as well as accuracy. We experimentally find that using BEV images of resolution 100x100 yields a good trade-off between computational cost and accuracy. We have included the above discussion in the *Training Details*.  Furthermore, we expanded the discussion of the relationships between the BEV size and the performances (runtime, memory statistics, and accuracy) in the revised appendix.

---

> ### Author Response · Authors · 2022-08-21
> **Response to Reviewer vD2G (4/7)**
>
> > * **Overall, the main question I feel should be answered is: when is desirable to enforce algorithmic priors instead of parameters?**
> > * **and a clearer answer to the question: "When are algorithmic priors a good idea?"**
>
> We believe the term ''Algorithmic Prior'' is firstly proposed by DPF and therefore please kindly allow us to cite a profound sentence from it first: *'Roboticists have captured problem structure in the form of algorithms, often combined with models of the specific task. **By making these algorithms differentiable and their models learnable, we can turn robotic algorithms into network architectures** ...... which we call algorithmic priors''*. We think the term "algorithmic prior" shares a similar idea to what we called **interpretability** in the manuscript. With the above definition of “algorithmic prior”, here is our answer to “when are algorithmic priors a good idea”:
>
> * When a mechanistic model can faithfully approximate a real-world system, adopting the mechanistic model and making it learnable, i.e., leveraging algorithmic prior, typically leads to better generalization performance. In this case, learnable parameters are only those that originally need empirical tunings when using a non-differentiable mechanistic model, e.g., the covariance matrices. This is also why we learn the covariance matrices in BEVO. As evidenced by our experiments, the mechanistic model of AV odometry satisfies the requirement of faithful approximation and thus using algorithmic prior is desired for better generalization.

---

> ### Author Response · Authors · 2022-08-21
> **Response to Reviewer vD2G (3/7)**
>
>
> > * **In addition, I feel that the main claim of this paper (rely more on algorithmic priors than parameters) is not sufficiently justified. It would be interesting to see results on more challenging datasets, possibly in 3D, where the assumption on the motion model would not always hold. If the paper only wishes to specialize in AV scenarios, then such choices must be made clear from the beginning (abstract/intro) and claims weakened.**
> > * **My recommendation is based on the disparity between the claims and the provided results.**
>
> Thank you for bringing this up! We have updated the introduction section that we focus on pose estimation of planar motion. We also discuss in the conclusion section that BEVO at this stage is incapable of handling general formulations in complete 3D motions, due to the difficulties in differentiating the numeric optimization. However, this should not deny the practicality of the BEV-based 2D motion estimation which has gained much attention recently in autonomous driving scenarios[7] [8]. Given the rationality of constructing VO model in BEV perspective, we believe BEVO is valuable to the AV scenarios. We have included the clarification in the *Abstract* and *Introduction* so that the introduction and the experiments are more aligned.
>
>
> To further verify that BEVO is generally applicable to AV scenarios, we conduct experiments on one additional AV dataset, KAIST's Complex Urban Dataset (CUD) [9].We train BEVO in scene ''Urban30'',''Urban31'',''Urban32'' and test the performance in scene "Urban33" as well as scene ''Urban34'',''Urban35''.. We have included these experiments in the revised appendix. Due to the tight timeline, we did not train the baselines in CUD to compare but demonstrate that the BEVO results in similar relative error on CUD compared to KITTI. We will further compare with other baselines in the camera-ready version.
>
> *[7]A Light-Weight Semantic Map for Visual Localization towards Autonomous Driving, Qin et al.*
>
> *[8]Avp-slam: Semantic visual mapping and localization for autonomous vehicles in the parking lot, Qin et al.*
>
> *[9]Complex Urban Dataset with Multi-level Sensors from Highly Diverse Urban Environments, Jeong et al.*

---

> ### Author Response · Authors · 2022-08-21
> **Response to Reviewer vD2G (2/7)**
>
> > * **A second interesting takeaway from this work is that traditional VIO methods, like VINS, perform terribly on KITTI. However, I find this somehow difficult to believe since, in the KITTI benchmark, most monocular systems are traditional VIO methods. Is this setup somehow different? What makes it challenging for them in the proposed evaluation?**
> > * **As mentioned before, the terrible performance of traditional VIO methods is difficult to interpret.**
>
> Thank you for mentioning this! In the manuscript, all results of VINS are borrowed from the paper: DeepVIO [5]. We carefully reassure that the evaluation settings and evaluation metrics in DeepVIO are the same as ours. The poor performance of traditional VIO methods could be due to the relatively low IMU rate and the uncritical synchronization in the dataset which cause covariances of the IMU data different from the statically calibrated, or datasheet parameters [6]. As tuning covariances by hand is difficult, traditional methods show weaker performance. The same reason is also given in SelfVIO[4] when explaining the poor performance of other traditional methods (OKVIS & ROVIO) in KITTI. Other traditional monocular VO methods that avoid the IMU covariance tuning such as ORB-M is also reported with poor performance since a mono-camera-only model is vulnerable to scaling problems. This, in our opinion, also account for why we perform better: we provide covariances that fit better for the low-frequency, uncritical synced IMU data. This claim can be validated in the ablation study, that BEVO achieves better performance than handcrafted covariances from experts. We have included the above analysis as well as the results for another traditional ORB-M in the experiments in the revision.
>
> *[5]DeepVIO: Self-supervised Deep Learning of Monocular Visual Inertial Odometry using 3D Geometric Constraints, Han et al.*
>
> *[6]Analysis and modeling of inertial sensors using Allan variance, Sheimy et al.*

---

> ### Author Response · Authors · 2022-08-21
> **Response to Reviewer vD2G (1/7)**
>
> Thank you for your valuable feedback. We address your concerns below, and have made revisions to the paper (highlighted in orange) based on your comments, which we believe has improved the paper. Please let us know if you have any remaining concerns or questions!
>
> > * **It is to be said, however, that similar ideas have been long investigated in the community (See [1], [2] or many others), but none really took off. This could be because learning so few parameters makes the algorithm strongly reliant on algorithm priors, which might be inappropriate in several real-world applications.**
> > * **Another weakness is the fact that previous work on deep state estimators (like [1],[2], and others) has not been sufficiently covered and discussed. In what does the approach differ? Why is it better?**
> > * **Also, I would like to see a discussion of previous methods.**
>
> Thank you for pointing out additional related work! We have updated the Introduction and discussed the suggested related works. Our method is indeed related to differentiable filtering methods (DF), including DPF [1], DKF [2], and DEKF [3]. In contrast to existing DF methods that combine interpretable and black-box modules, our method is fully interpretable and leads to better generalization performance. We now elaborate on the comparison in the following.
>
> Existing DF methods and our proposed approach all deal with sensor fusion, which consists of a sensor model and fusion mechanism. Compared to a fully black-box baseline (e.g., LSTM), DF methods achieve better generalization performance by modeling fusion mechanism with interpretability, while keeping the sensor model as a black box. However, when sensor models are equipped with some interpretability, e.g., SelfVIO [4] with perspective geometry, the black-box sensor model in DF methods might become the bottleneck for achieving superior generalization performance. In other words, we consider that the black-box sensor model might be the reason why these methods did not really take off.
>
> Following this idea, we hypothesize that better generalization performance can be achieved when both fusion and sensor models are fully interpretable. This is actually a traditional VIO method with mechanistic models. We agree that this method is strongly reliant on the correctness of the mechanistic model, yet the mechanistic model is appropriate for VIO application with decades of community efforts. We consider that one of the problems preventing the traditional VIO methods is handcrafted parameters in covariances, which we learn from data in this work. Therefore, we focus on the differentiable sensor model to back-propagate the loss to covariance. As shown in TABLE 1, Figure 4 in the manuscript and TABLE 1 in the appendix, the hypothesis can be validated. It means our better performance compared to existing DF methods mainly comes from using a fully interpretable sensor model, i.e., the differentiable phase correlation model.
>
> *[1]Differentiable Particle Filters: End-to-End Learning with Algorithmic Priors, Jonschkowski et al.*
>
> *[2]Deep Kalman Filters, Krishnan et al.*
>
> *[3]Deep Robust Kalman Filter, Di-Castro et al.*
>
> *[4]Self-supervised deep monocular Visual–Inertial Odometry and depth estimation, Almalioglu et al.*

---

> > ### Comment · Reviewer_vD2G · 2022-08-26
> > **Answer to rebuttal**
> >
> > I thank the author for the detailed rebuttal and the additional experiments. While I agree with the other comments, I am not convinced by the following argument __we consider that the black-box sensor model might be the reason why these methods did not really take off.__
> > There is no evidence to back up this claim. Therefore, the paragraph starting at line 45 of the new manuscript does not seem justified. I see the proposed method as a middle-ground between traditional VIO and the aforementioned baselines. Therefore, to understand if the merit comes from eliminating the black-box sensor model, a more thorough experimental study is needed.

---

> > > ### Author Response · Authors · 2022-08-27
> > > **Reply to the answer**
> > >
> > > **Comment:**
> > >
> > > > * **I am not convinced by the following argument *“we consider that the black-box sensor model might be the reason why these methods did not really take off.”* There is no evidence to back up this claim.**
> > > > * **Therefore, to understand if the merit comes from eliminating the black-box sensor model, a more thorough experimental study is needed.**
> > >
> > > Thank you for the valuable feedback! We agree that the statement is not properly backed by any experiments and therefore, we have conducted two more ablation studies to prove the better performance of the mechanistic model than the black-box sensor model. We have included the analysis in the latest revised manuscript and have also uploaded a newer version of the code.
> > >
> > > Following the methodology in [1] and [2], we eliminate the mechanistic sensor model, deep phase correlation, and replace with ResNet blocks and train it to retrieve the translations and rotation between two BEV images.
> > >
> > > 1. In the first experiment, To control the variables, we do not train the DUKF and the covariances are given by the original BEVO, which is trained end to end with DUKF and DPC. With the DUKF set, the BEV images presented to the ResNet are in the same condition as those fed into DPC and we train the ResNet individually. The experiment denoted as ''Res'' in Table. 3 shows that when we replace the DPC with the ResNet, results in the train/val set are comparable to the original BEVO. However, for generalization in seq.09 and seq.10, the performance is greatly diminished. To make sure the ResNet is not overfitted, we test the performance with multiple checkpoints and choose the best one. In addition, we notice that this combination is still better than the method without DUKF (w/o DUKF in Table. 3), showing the importance of the interpretable fusion model.
> > > 2. The second experiment is conducted in an end to end manner. The DUKF is not previously trained, and is trained with the ResNet altogether. However, the training process is hindered, and we believe that the network can hardly converge within the remaining rebuttal time period. This phenomenon is quite common for end to end methods with multiple trainable parts, and in our case, it is also explainable: without a good DUKF, the BEV images are not orthogonal, which might lead to a bad convergence of the ResNet.
> > >
> > > By conducting this experiment, we arrive at a result that by replacing the black box sensor model with the interpretable mechanistic one, such as DPC, the method shows better performance. Similar results are shown in [3] [4] which find that considering geometry in the sensor model improves the performance.
> > >
> > >
> > > *[1]Differentiable Particle Filters: End-to-End Learning with Algorithmic Priors, Jonschkowski et al.*
> > >
> > > *[2]Backprop KF: Learning Discriminative Deterministic State Estimators, Haarnoja et al.*
> > >
> > > *[3]DF-VO: What Should Be Learnt for Visual Odometry? Zhan et al.*
> > >
> > > *[4]Visual odometry revisited: What should be learnt? Zhan et al.*
> > >
> > >
> > > **Zip File:**
> > >
> > > /attachment/577f8010835d8fc6d961ddf3a87ee5f0443e6182.zip

---

### Official Review · Reviewer_Me8P · 2022-08-01

**Originality:** Very Good
**Technical Quality:** Very Good
**Clarity Of Presentation:** Excellent
**Impact:** 4

**Recommendation:**

Strong Accept: I recommend accepting the paper and will argue for my recommendation even if other reviewers hold a different opinion.

**Summary:**

This paper proposes a novel VIO method which estimates 5 DoF poes (with the elevation fixed) from monocular camera and IMU. In particular the proposed method adopt Unscented Kalman Filter (UKF) which is natually differentiable for the state estimation. With the aid of filtered pitch and roll using UKF, the front-view images can be converted to bird eye view images which are then used for pose estimation with Differentiable Phase Correlation. The whole method is differentiable and can be backpropagated using ground-truth pose supervision to update the covariance matrices. Evaluation results show the the proposed method achieves competitive results on real-world KITTI dataset for odometry, CARLA and AeroGround datasets for map-based localization. The authors also conducted ablation studies on the learned covariance matrices.

**Issues:**

It would be great if the authors could address my concers in the *Strengths And Weaknesses* section.

**Quality Of The Limitations Section:**

Limitations are addressed clearly

**Reviewer Expertise:**

4: The reviewer is confident but not absolutely certain that the evaluation is correct

**Robotics Focus:**

Highly relevant to robotics but no hardware experiments

**Strengths And Weaknesses:**

## Strengths
- The paper is very well written and one can read very smoothly.
- The authors make use of the differentiable nature of Unscented Kalman Filter (UKF) and proposes to learn the covariance matrices in a end-to-end manner to capture the motion and measurement noise distribution. This way it retains the interpretability of the pipeline and also leverage the power of data-driven approach to learn the otherwise hand-tuned covariance matrices.
-  The estimated roll and pitch from UKF is integrated seamlessly to the bird eye view projection followed by camera pose estimation using Differentiable Phsace Correlation. This enables the differentiability of the whole piple with reasonable assumptions.
- The results show very competitive results on KITTI for camera pose estimations. Further, the authors also show its effectiveness to apply it for map-based localization on CARLA and AeroGround datasets.
- The authors provide the code in the supplement.

## Weaknesses
- Altough the whole method only learns 4 parameters, it is a bit confusing why it needs 2 RTX 3090 to train.
- No runtime and memory statistics are provided.
- The authors did not show cross dataset generalization evaluation. It would be interesting to see if the covariances learned from one dataset/sensor can transfer to other datasets.

**Summary Of Recommendation:**

The authors propose a quite novel approach for 5 DoF VIO which leverages differentialble UKF filter and BEV projection. The learned covarinances are shown to be very effective and I believe such an approach could have a big impact to the community.

---

> ### Author Response · Authors · 2022-08-21
> **Response to Reviewer Me8P (3/3)**
>
> > **The authors did not show cross dataset generalization evaluation. It would be interesting to see if the covariances learned from one dataset/sensor can transfer to other datasets.**
>
> Thank you for mentioning this! Benefiting from the mechanistic model, we can gain a concrete understanding of what kind of generalization BEVO can achieve. Since the covariance matrix learned is critically related to the IMU sensor,  a single set of covariance parameters is unlikely to generalize across different IMU sensors, as it is not grounded by mechanistic model. When two datasets share the same sensor, we expect the covariances learned from one dataset to generalize well on the other. However, it is pretty hard to find two different datasets with the same IMU sensor. Therefore, we did not test the generalization ability across datasets, but evaluated on testing sequences 09 \& 10 of KITTI as a generalization experiment across scenes with the same sensor.
>
> Note that it is convenient to train BEVO on another dataset captured by a different sensor. Therefore, to further verify cross-scene generalization ability, we added an experiment on KAIST's Complex Urban Dataset (CUD) in the revised appendix, in which we train BEVO in scene ''Urban30'',''Urban31'',''Urban32'' and test the performance in scene "Urban33" as well as scene ''Urban34'',''Urban35''. Note these sequences are collected by the same sensor, but from different locations. We have included these experiments in the revised appendix. Due to the tight timeline of the rebuttal, we did not train other baselines in CUD to compare but only demonstrated the performance of BEVO, however, we will further compare with other baselines in the camera-ready version.
>
>
>
> As a conclusion, following the advice, we carefully revised the manuscript with:
>
> * We have included the runtime, memory statistics and the size of the BEV in the *Training Details*.
> * We expanded the discussion of the relationships between the BEV size and the performances (runtime, memory consumption, and accuracy) in the appendix.
> * We added more experiments on KAIST's Complex Urban Dataset in the appendix to justify the cross-scene generalization ability.

---

> ### Author Response · Authors · 2022-08-21
> **Response to Reviewer Me8P (2/3)**
>
> > **No runtime and memory statistics are provided.**
>
> Thanks for this advice. In all of our experiments, the inference time for one motion is 21ms, and the memory usage is 500MB. This is obtained using BEV images with a resolution of 100x100. Intuitively, the size of the BEV images affects runtime, memory usage as well as accuracy. We experimentally find that using BEV images of resolution 100x100 yields a good trade-off between computational cost and accuracy. We have included the above discussion in the *Training Details*.  Furthermore, we expanded the discussion of the relationships between the BEV size and the performances (runtime, memory statistics, and accuracy) in the revised appendix.

---

> ### Author Response · Authors · 2022-08-21
> **Response to Reviewer Me8P (1/3)**
>
> Thank you for your valuable feedback. We address your concerns below, and have made revisions to the paper (highlighted in orange) based on your comments, which we believe has improved the paper. Please let us know if you have any remaining concerns or questions!
>
> > **Although the whole method only learns 4 parameters, it is a bit confusing why it needs 2 RTX 3090 to train.**
>
> Thank you for catching this! The framework is indeed lightweight with 500MB for forward and 1.8GB for backward, and does not need the full GPU memory provided by RTX 3090 to run. We apologize for the confusion we introduced in the manuscript, we intentionally mean that the server we conduct experiments on has 2 RTX 3090, but we use only one of them for training and inference in each experiment. We have revised the manuscript to avoid this confusion.
> Additionally, we trained the network by sequence for each iteration (one sequence includes 10 camera frames and the IMU data within the corresponding time) so a better GPU is welcomed.

---

### Author Response · Authors · 2022-08-21
**Revised Manuscript and Appendix**

**Comment:**

We thank meta-reviewer and all the reviewers for all the insightful advice.We have made revisions to the paper (highlighted in orange) based on all the comments, which we believe has improved the paper. Please let us know if you have any remaining concerns or questions!

**Zip File:**

/attachment/7159dbbeed26de4892ea7ac653c93f217510e058.zip

---

### Meta-Review · Area_Chair_i2xg · 2022-08-14

**Recommendation:** Accept (Poster)
**Confidence:** 4

**Metareview:**

The paper proposes a differentiable approach for monocular VIO estimation based on BEV, without relying on deep neural networks. The reviewers find the paper well written and the idea of using BEV to be interesting. This paper received highly mixed reviews. The major concerns raised by the reviewers include empirical evaluations of the model interpretability, justification for relying on algorithmic priors than parameters, results on more challenging datasets, positioning this work with respect to existing work on deep state estimators, and clarifications regarding the claims made, among others. Most of the concerns raised by the reviewers have been thoroughly addressed in the rebuttal. I thank the authors for the engaging discussions during the rebuttal. Some minor concerns still exist. Nevertheless, I agree with the reviewers that the paper is an interesting contribution.

**Best Paper Nomination:**

No

---

> ### Author Response · Authors · 2022-08-21
> **Response to Meta Reviewer i2xg (6/6)**
>
> > **Runtime should be compared** (from reviewer Me8P, vD2G)
>
> In all of our experiments, the inference time for one motion is 21ms, and the memory usage is 500MB. This is obtained using BEV images with a resolution of 100x100. Intuitively, the size of the BEV images affects runtime, memory usage as well as accuracy. We experimentally find that using BEV images of resolution 100x100 yields a good trade-off between computational cost and accuracy. We have included the above discussion in the *Training Details*.  Furthermore, we expanded the discussion of the relationships between the BEV size and the performances (runtime, memory statistics, and accuracy) in the revised appendix.

---

> ### Author Response · Authors · 2022-08-21
> **Response to Meta Reviewer i2xg (5/6)**
>
> > **Positioning this work with respect to existing work on deep state estimators, clarifications regarding the claims made, among others.** (from reviewer vD2G)
>
> Our method is indeed related to differentiable filtering methods (DF), including DPF [4], DKF [5], and DEKF [6]. In contrast to existing DF methods that combine interpretable and black-box modules, our method is fully interpretable and leads to better generalization performance. We now elaborate on the comparison in the following.
>
> Existing DF methods and our proposed approach all deal with sensor fusion, which consists of a sensor model and fusion mechanism. Compared to a fully black-box baseline (e.g., LSTM), DF methods achieve better generalization performance by modeling fusion mechanism with interpretability, while keeping the sensor model as a black box. However, when sensor models are equipped with some interpretability, e.g., SelfVIO [4] with perspective geometry, the black-box sensor model in DF methods might become the bottleneck for achieving superior generalization performance. In other words, we consider that the black-box sensor model might be the reason why these methods did not really take off.
>
> Following this idea, we hypothesize that better generalization performance can be achieved when both fusion and sensor models are fully interpretable. This is actually a traditional VIO method with mechanistic models. We agree that this method is strongly reliant on the correctness of the mechanistic model, yet the mechanistic model is appropriate for VIO application with decades of community efforts. We consider that one of the problems preventing the traditional VIO methods is handcrafted parameters in covariances, which we learn from data in this work. Therefore, we focus on the differentiable sensor model to back-propagate the loss to covariance. As shown in TABLE 1, Figure 4 in the manuscript and TABLE 1 in the appendix, the hypothesis can be validated. It means our better performance compared to existing DF methods mainly comes from using a fully interpretable sensor model, i.e., the differentiable phase correlation model.
>
> *[4]Differentiable Particle Filters: End-to-End Learning with Algorithmic Priors, Jonschkowski et al.*
>
> *[5]Deep Kalman Filters, Krishnan et al.*
>
> *[6]Deep Robust Kalman Filter, Di-Castro et al.*

---

> ### Author Response · Authors · 2022-08-21
> **Response to Meta Reviewer i2xg (4/6)**
>
> > **Experiments for cross-dataset generalization should be added** (from reviewer Me8P)
>
> Benefiting from the mechanistic model, we can gain a concrete understanding of what kind of generalization BEVO can achieve. Since the covariance matrix learned is critically related to the IMU sensor,  a single set of covariance parameters is unlikely to generalize across different IMU sensors, as it is not grounded by mechanistic model. When two datasets share the same sensor, we expect the covariances learned from one dataset to generalize well on the other. However, it is pretty hard to find two different datasets with the same IMU sensor. Therefore, we did not test the generalization ability across datasets, but evaluated on testing sequences 09 \& 10 of KITTI as a generalization experiment across scenes with the same sensor.
>
> Note that it is convenient to train BEVO on another dataset captured by a different sensor. Therefore, to further verify cross-scene generalization ability, we added an experiment on KAIST's Complex Urban Dataset (CUD) in the revised appendix, in which we train BEVO in scene ''Urban30'',''Urban31'',''Urban32'' and test the performance in scene "Urban33" as well as scene ''Urban34'',''Urban35''. Note these sequences are collected by the same sensor, but from different locations. We have included these experiments in the revised appendix. Due to the tight timeline of the rebuttal, we did not train other baselines in CUD to compare but only demonstrated the performance of BEVO, however, we will further compare with other baselines in the camera-ready version.

---

> ### Author Response · Authors · 2022-08-21
> **Response to Meta Reviewer i2xg (3/6)**
>
> > **Results on more challenging datasets (potentially 3D)** (from reviewer vD2G)
>
> We have updated the introduction section that we focus on pose estimation of planar motion. We also discuss in the conclusion section that BEVO at this stage is incapable of handling general formulations in complete 3D motions, due to the difficulties in differentiating the numeric optimization. However, this should not deny the practicality of the BEV-based 2D motion estimation which has gained many attentions recently in autonomous driving scenarios[2] [3]. Given the rationality of constructing VO model in BEV perspective, we believe BEVO is valuable to the AV scenarios. We have included the clarification in the *Abstract* and *Introduction* so that the introduction and the experiments are more aligned.
>
>
> To further verify that BEVO is generally applicable to AV scenarios, we conduct experiments on one additional AV dataset, KAIST's Complex Urban Dataset (CUD). We train BEVO in scene ''Urban30'',''Urban31'',''Urban32'' and test the performance in scene "Urban33" as well as scene ''Urban34'',''Urban35''.. We have included these experiments in the revised appendix. Due to the tight timeline, we did not train the baselines in CUD to compare but demonstrate that the BEVO results in similar relative error on CUD compared to KITTI. We will further compare with other baselines in the camera-ready version.
>
> *[2]A Light-Weight Semantic Map for Visual Localization towards Autonomous Driving, Qin et al.*
>
> *[3]Avp-slam: Semantic visual mapping and localization for autonomous vehicles in the parking lot, Qin et al.*

---

> ### Author Response · Authors · 2022-08-21
> **Response to Meta Reviewer i2xg (2/6)**
>
> > **Justification for relying on algorithmic priors than parameters** (from reviewer vD2G)
>
> We believe the term ''Algorithmic Prior'' is firstly proposed by DPF and therefore please kindly allow us to cite a profound sentence from it first: *'Roboticists have captured problem structure in the form of algorithms, often combined with models of the specific task. **By making these algorithms differentiable and their models learnable, we can turn robotic algorithms into network architectures** ...... which we call algorithmic priors''*. We think the term "algorithmic prior" shares a similar idea to what we called **interpretability** in the manuscript. With the above definition of “algorithmic prior”, here is our answer to “when are algorithmic priors a good idea”:
>
> * When a mechanistic model can faithfully approximate a real-world system, adopting the mechanistic model and making it learnable, i.e., leveraging algorithmic prior, typically leads to better generalization performance. In this case, learnable parameters are only those that originally need empirical tunings when using a non-differentiable mechanistic model, e.g., the covariance matrices. This is also why we learn the covariance matrices in BEVO. As evidenced by our experiments, the mechanistic model of AV odometry satisfies the requirement of faithful approximation and thus using algorithmic prior is desired for better generalization.

---

> ### Author Response · Authors · 2022-08-21
> **Response to Meta Reviewer i2xg (1/6)**
>
> We thank the Area Chair i2xg for the meta review, comments, suggestions and recommendations as well as the thorough summary. We first briefly address the common concerns of the reviewers here, and please kindly let us know if you have any remaining concerns or questions!
>
> > **Empirical evaluations of the model interpretability** (from reviewer 3Som)
>
> We would like to clarify the interpretability of the method from our point of view. Citing from [1]: *"Interpretable ML focuses on designing models that are inherently interpretable; whereas explainable ML tries to provide post hoc explanations for existing black box models"*, we would like to reclaim that we are not trying to explain a black box but to design a model that is inherently interpretable. If results are learned using black-box components like FCN, it is uninterpretable, let alone design non-trainable components to replace it. However, our model does not contain such uninterpretable components, and the four learnable parameters have a user-understandable meaning: the covariances of IMU measurements.
>
> If the reviewer can kindly refer to the ablation study in the manuscript, we have actually conducted experiments eliminating the learnable parameters in the method to restore the method to a mechanistic one. These parameters have specific definitions in the mechanistic model and are given by data sheet, calibration, or expert experiences in traditional methods, which means the users do have a concrete understanding of the parameters. In the ablation study, experts designed a set of parameters based on [2] to replace the learned one, and the method still works, but could not find a better one to achieve competitive performance to the data-driven parameters. Also, taking the poor performance of traditional methods VINS and ORB-M in KITTI for example, the low IMU frequency and the uncritical sync of the IMU make it hard for experts to design a perfect parameter.
>
> * Therefore we argue that by combining interpretable mechanistic models with end-to-end learning of a few parameters, the inherent interpretability is not diminished, but even will thrive with more suitable parameters. This can be justified by the ablation study.
>
> [1]The Unscented Kalman Filter, E.A et al.